# Long Non-Coding RNA-Cardiac-Inducing RNA 6 Mediates Repair of Infarcted Hearts by Inducing Mesenchymal Stem Cell Differentiation into Cardiogenic Cells through Cyclin-Dependent Kinase 1

**DOI:** 10.3390/ijms25063466

**Published:** 2024-03-19

**Authors:** Xiaotian Cui, Hui Dong, Shenghe Luo, Bingqi Zhuang, Yansheng Li, Chongning Zhong, Yuting Ma, Lan Hong

**Affiliations:** Department of Physiology and Pathophysiology, College of Medicine, Yanbian University, Yanji 133002, China; ajongcs@outlook.com (X.C.); qingjiu30@gmail.com (H.D.); 2021010886@ybu.edu.cn (S.L.); 15957661009@163.com (B.Z.); ysli@ybu.edu.cn (Y.L.); 13944948637@163.com (C.Z.); 15981188912@163.com (Y.M.)

**Keywords:** LncRNA-CIR6, myocardial infarction, BMSCs, hUMSCs, CDK1

## Abstract

This study aims to investigate the induction effect of LncRNA-CIR6 on MSC differentiation into cardiogenic cells in vitro and in vivo. In addition to pretreatment with Ro-3306 (a CDK1 inhibitor), LncRNA-CIR6 was transfected into BMSCs and hUCMSCs using jetPRIME. LncRNA-CIR6 was further transfected into the hearts of C57BL/6 mice via 100 μL of AAV9-cTnT-LncRNA-CIR6-ZsGreen intravenous injection. After three weeks of transfection followed by AMI surgery, hUCMSCs (5 × 10^5^/100 μL) were injected intravenously one week later. Cardiac function was evaluated using VEVO 2100 and electric mapping nine days after cell injection. Immunofluorescence, Evans blue-TTC, Masson staining, FACS, and Western blotting were employed to determine relevant indicators. LncRNA-CIR6 induced a significant percentage of differentiation in BMSCs (83.00 ± 0.58)% and hUCMSCs (95.43 ± 2.13)% into cardiogenic cells, as determined by the expression of cTnT using immunofluorescence and FACS. High cTNT expression was observed in MSCs after transfection with LncRNA-CIR6 by Western blotting. Compared with the MI group, cardiac contraction and conduction function in MI hearts treated with LncRNA-CIR6 or combined with MSCs injection groups were significantly increased, and the areas of MI and fibrosis were significantly lower. The transcriptional expression region of LncRNA-CIR6 was on Chr17 from 80209290 to 80209536. The functional region of LncRNA-CIR6 was located at nucleotides 0–50/190–255 in the sequence. CDK1, a protein found to be related to the proliferation and differentiation of cardiomyocytes, was located in the functional region of the LncRNA-CIR6 secondary structure (from 0 to 17). Ro-3306 impeded the differentiation of MSCs into cardiogenic cells, while MSCs transfected with LncRNA-CIR6 showed a high expression of CDK1. LncRNA-CIR6 mediates the repair of infarcted hearts by inducing MSC differentiation into cardiogenic cells through CDK1.

## 1. Introduction

Acute myocardial infarction (AMI) is a serious cardiovascular disease [1]. The proliferation of cardiomyocytes during normal aging cannot reverse the damage caused by acute myocardial infarction, making it difficult for cardiomyocytes to regenerate. This difficulty leads to abnormal recovery of cardiac function [2,3]. However, cell therapy using stem cells can effectively compensate for the loss of cardiac myocytes, overcoming the challenges of traditional medical methods [4].

Embryonic stem cells (ESCs) and induced pluripotent stem cells (iPSCs) have been used in clinical trials for cell therapies such as cristae injuries, macular degeneration, and Type 1 diabetes [5,6,7]. However, as the application of stem cell therapy to cardiac diseases has not been standardized, it has not yet been introduced into clinical treatment. It is known that the use of stem cells in experimental models shows improvement in cardiac function and the promotion of angiogenesis, along with the secretion of biologically active paracrine molecules that contribute to cardiac protection [8]. The transplantation of embryonic stem cells modified with CREG has been shown to improve cardiac function in mice after myocardial infarction [9]. Additionally, a wealth of preclinical and early clinical data suggests the safety, feasibility, and early efficacy of stem cell transplantation [10]. Studies have found that bone marrow-derived mononuclear cells, in the treatment of acute myocardial infarction, quickly transitioned to phase III clinical trials [11,12].

However, clinical transplantation faces many limitations, including the immature characteristics of the heart, long-term implantation issues, transplantation-related arrhythmias, immunogenicity, and the risk of tumorigenesis [13]. Early studies indicated that autologous skeletal muscle myoblast transplantation in phase I clinical trials could lead to sustained ventricular tachycardia [14]. Furthermore, concerns have been raised about the effectiveness of granulocyte colony-stimulating factor (G-CSF) in mobilizing stem/progenitor cells in clinical trials for coronary artery disease patients and its safety concerning its association with arterial restenosis and plaque instability [15,16]. These concerns underscore the necessity for further exploration of this emerging and exciting therapeutic approach for cardiovascular diseases.

Compared with induced iPSCs and ESCs, mesenchymal stem cells (MSCs), as the seed cells of regenerative medicine and tissue engineering, have many advantages, including low immune rejection, easy preparation, the large size of MSCs, and a lack of ethical influence [17,18]. Studies have found that MSCs can promote myocardial regeneration after myocardial infarction [19]. Transplanting MSCs not only compensates for lost heart cells but also regulates immune factors [20]. Cardiac MSCs reduce inflammation and ameliorate myocardial infarction in rats with ischemia-reperfusion injuries [21]. MSCs and their secreted products decrease the gene expression of pro-inflammatory cytokines and interleukin in cardiomyocytes, while the expression of antioxidant gene superoxide dismutase is upregulated [22,23]. Therefore, understanding how to regulate the proliferation and differentiation of MSCs and how to use mesenchymal stem cells to cure diseases is an urgent need in regenerative science.

The study of long non-coding RNA (LncRNA) in stem cells is an active and fruitful field [24]. In recent years, an increasing number of studies have shown that lncRNA plays a crucial role in the self-renewal, differentiation, reprogramming, and other processes of stem cells [24,25,26]. As an intracellular gene regulator, LncRNA can mediate the development of mesenchymal stem cells [25,26]. For example, LncRNA H19 can repair damaged myocardium by improving the viability of MSCs and the angiogenic capacity of endothelial cells [27]. Similarly, LncRNA TCF7 may maintain the cellular viability and stem cell viability of MSCs by activating the Wnt pathway [28].

The lncRNA-CIR6 from the human heart was initially identified by the Center for Regenerative Medicine at Texas A&M University. In non-cardiomyocytes, such as mouse embryonic stem cells, mouse and human iPSCs, and mouse embryonic fibroblasts, LncRNA-CIR6 can be induced to differentiate into cardiomyocytes expressing cTnT in vitro [29,30,31]. However, it is not clear whether LncRNA-CIR6 can also induce MSCs.

## 2. Results

### 2.1. Effect of LncRNA-CIR6 on the Induced Differentiation of BMSCs and hUCMSCs into Cardiomyogenic Cells

The morphological changes of BMSCs and hUCMSCs during differentiation after transfection with empty plasmid or LncRNA-CIR6 plasmid were observed by the inverted optical microscope. BMSCs and hUCMSCs in the Vehicle group mainly showed a spindle shape. However, these cells in the LncRNA-CIR6 group gradually showed morphological characteristics similar to cardiomyocytes, and some cells even formed a structural arrangement similar to rhabdomytic muscle (Figure 1A).

To further validate the potential of LncRNA-CIR6 for inducing the differentiation of BMSCs and hUCMSCs into cardiomyogenic cells, we examined the cardiomyocyte-specific marker cTNT through immunofluorescence and Western blot analyses. The results obtained on the 10th day post-transfection with LncRNA-CIR6 revealed green fluorescence in both MSCs (with the empty plasmid and LncRNA-CIR6 plasmid carrying GFP-Green fluorescent protein), indicating successful transfection. Additionally, the cells exhibited clear expression of red fluorescence-labeled cTNT protein. In contrast, BMSCs and hUCMSCs transfected with empty plasmids exhibited only green fluorescence (Figure 1B). Furthermore, the Western blot analysis demonstrated a significant increase in cTNT expression in MSCs transfected with LncRNA-CIR6, while the opposite was observed in the vehicle group (Figure 1C). These findings collectively indicate that LncRNA-CIR6 induces the differentiation of both MSCs into cardiomyogenic cells.

Additionally, we assessed the efficiency of LncRNA-CIR6 in inducing MSCs to differentiate into cardiomyogenic cells by measuring the fluorescence intensity of APC through flow cytometry. The results showed that LncRNA-CIR6 induced an average conversion rate of (83.00 ± 0.5774)% in BMSCs and (95.43 ± 2.1300)% in hUCMSCs (Figure 1D). These results underscore the capability of LncRNA-CIR6 to induce MSCs to differentiate into cardiogenic cells.

### 2.2. The Effects of LncRNA-CIR6 or Combined hUMSCs on Heart Function and Cardioprotection In Vivo in Mice with Myocardial Infarction

The study assessed the heart’s contraction and conduction functions through echocardiography and Electrical Mapping. Cardiac function before and after myocardial infarction (MI) was compared using the VEVO 2100 imaging system, as depicted in Figure 2A. Preoperatively, the left ventricular contraction function was robust, and the left ventricular wall exhibited normal characteristics. Postoperatively, the left ventricular anterior wall showed thinning, reduced or absent segmental motion, impaired contraction function, tissue remodeling, and other manifestations.

However, it was observed that the cardiac function in both the LncRNA-CIR6 group and LncRNA-CIR6 + hUCMSCs group was superior to the MI group. Furthermore, based on the ultrasound evaluation results, both LVID,s and LVID,d significantly increased in the MI group, while LVEF and LVFS significantly decreased (Figure 2B). In contrast, the LncRNA-CIR6 group and LncRNA-CIR6 + hUCMSCs group exhibited a reduction in LVID,s and LVID,d, accompanied by an increase in LVEF and LVFS. These findings suggest that the administration of LncRNA-CIR6 or a combined injection with hUCMSCs can improve the heart’s contraction function.

The results from the Electrical Mapping measurements indicate that, compared to the MI group, both the LncRNA-CIR6 group and the LncRNA-CIR6 + hUCMSCs group exhibited a reduction in conduction time and dispersion values, accompanied by an increase in conduction velocity, suggesting an improvement in conduction capability (Figure 2C–F). Notably, the combined therapy of LncRNA-CIR6 and hUCMSCs demonstrated a superior effect on conduction function following myocardial infarction (Figure 2C–F).

To delve deeper into the protective effects of LncRNA-CIR6 or the combined treatment with hUMSCs on myocardial infarction, postoperative mouse myocardial infarction and fibrosis were examined using fluorescence, Evans Blue-TTC staining, and Masson staining. The findings revealed a substantial reduction in infarct size in mice from both the LncRNA-CIR6 group and the LncRNA-CIR6 + hUCMSCs group compared to the MI group, as demonstrated by fluorescence (Figure 2G). This was consistent with Evan Blue-TTC staining results, displaying an increase in the red area and a decrease in the blue area (Figure 2H). Additionally, histological analysis using Masson staining indicated a notable decrease in the myocardial fibrosis area in both the LncRNA-CIR6 group and the LncRNA-CIR6 + hUCMSCs group (Figure 2I). Importantly, when compared to individual treatments, the combined therapy of LncRNA-CIR6 and hUMSCs significantly inhibited both the myocardial infarction area and the degree of fibrosis (Figure 2G–I).

In summary, the application of LncRNA-CIR6 or its combination with hUMSCs cannot only enhance post-infarction myocardial contraction and conduction functions but also mitigate infarct size and fibrosis.

### 2.3. Role of CDK1 in LncRNA-CIR6 Induced Differentiation of MSCs into Cardiogenic Cells

To further explore the mechanism underlying the induction of MSCs by LncRNA-CIR6, we conducted a comprehensive biological information analysis. Using Annolnc2, we identified the transcriptional expression region of LncRNA-CIR6 on the reverse strand of chromosome 17 in the human genome. The structure of this transcript closely resembles that of the known transcript N-sulfonylglucose sulfate hydrolase (Figure 3A). Subsequently, we obtained the secondary structure (Figure 3B) of LncRNA-CIR6 from database analysis and the mountain chart predicted by the database (Figure 3C). We interpreted this information based on the free energy value of LncRNA molecules, where a lower free energy value indicates higher stability and functionality, rendering them more likely to be associated with diseases. Therefore, based on the mountain chart of LncRNA-CIR6 (Figure 3C), we predicted that the functional region spans nucleotides 0–50/190–255 (highlighted in red in Figure 3B).

CLIP-seq data revealed that LncRNA-CIR6 interacted with 28 proteins across nine cell types. Among the 28 predicted proteins, six had more than 10 binding sites with LncRNA-CIR6, and all binding sites were in the functional region sequence of LncRNA-CIR6. Simultaneously, these predicted proteins were all related to cardiovascular system diseases, as depicted in Area 6 in Figure 3D.

Further analysis revealed that among these six proteins, CDK1 was the only one associated with the proliferation and differentiation of cardiomyocytes [32]. Moreover, the binding site of CDK1 in LncRNA-CIR6 was located between 0 and 17 of LncRNA-CIR6, belonging to the stable functional region (highlighted by the red box in Figure 3E). To predict whether CDK1 was involved in the induction of MSC differentiation into cardiogenic cells by LncRNA-CIR6, Western blot experiments were performed. CDK1 was found to be highly expressed in the LncRNA-CIR6-transfected group (Figure 3F), confirming our initial prediction of the association between LncRNA-CIR6 and CDK1.

Meanwhile, MSCs transfected with the CDK1 inhibitor Ro-3306 did not exhibit the heart-specific protein cTNT despite LncRNA-CIR6 transfection. This suggests that Ro-3306 blocked the effect of LncRNA-CIR6 on inducing MSCs to differentiate into cardiogenic cells (Figure 3G), further affirming the hypothesis that CDK1 may play a role in the LncRNA-CIR6-induced differentiation of MSCs into cardiogenic cells.

## 3. Discussion

Long non-coding RNAs (LncRNAs) have been confirmed to play crucial roles in the pluripotency of stem cells and cardiac differentiation [33]. LncRNAs constitute an essential part of the transcriptional network in stem cells [34]. Among them, the knockdown or overexpression of LncRNAs can reciprocally influence pluripotency transcription factors, thereby regulating the pluripotent state and lineage specificity of stem cells [35].

Research has identified LncDACH1 as a negative regulator of cardiac regeneration [33]. Its silencing enhances the proliferative potential of myocardial cells, reduces infarct size, and improves cardiac function [36]. Additionally, LncRNAs play a critical role in the process of cellular reprogramming, especially in somatic cell reprogramming [37]. For instance, LncRNA-Bvht is a known LncRNA associated with heart formation during embryonic development, and its expression can prompt embryonic stem cells (ESCs) to differentiate into cardiac muscle cells [38,39]. These findings provide new insights into the regulatory role of LncRNAs in the proliferation and differentiation of cardiac muscle cells, offering potential therapeutic strategies for research in cardiac regenerative medicine.

CIR was initially discovered by the Biomedical Institute for Regenerative Research (BIRR) at Texas A&M University-Commerce [29,30]. The researchers identified a unique RNA in the endoderm of normal axolotl hearts that could rescue the hearts of mutant axolotl embryos. This RNA also had the ability to induce non-functional cardiomyocytes in mutant axolotl hearts to transform into cardiomyocytes with normal myofibrillar fibers. The researchers named this RNA Myocardial-Inducing RNA (MIR) [30]. Further investigations revealed that three RNAs from the human fetal heart could also rescue mutant axolotl embryonic hearts. These RNAs were named Cardiac-Inducing RNA (CIR) 6, 30, and 291 [31,40,41]. Through an analysis of the secondary structure, it was found that CIR6 had a remarkably similar secondary structure to MIR. Subsequent studies confirmed that CIR6 does not possess messenger RNA properties or functions, nor does it function as Ligand RNA [31,40,41]. This suggests that CIR6 is a non-coding RNA. CIR6 has been identified as a Long non-coding RNA (LncRNA) [31,41].

LncRNA-CIR6 is a long non-coding RNA specifically expressed in the human fetal heart [41]. Its crucial role is evident in inducing pluripotent stem cells to form cardiomyocytes in both mouse embryos and humans [31]. In addition to expressing markers like cardiac troponin T (cTNT), LncRNA-CIR6 also exhibits cardiac-specific contractile protein markers, including original myosin and α-myosin [31]. Notably, LncRNA-CIR6 possesses an evolutionarily conserved secondary structure, enabling it to promote the differentiation of non-muscle cells into cardiomyocytes [31]. Consequently, we successfully induced the differentiation of MSCs into cardiogenic cells through the transfection of LncRNA-CIR6. The induction of LncRNA-CIR6 significantly improved the efficiency of cardiogenic cell differentiation in MSCs, leading to the induction of the cardiac-specific marker cTNT. These findings suggest that LncRNA-CIR6 may play a crucial role in signaling pathways associated with cardiogenic cell differentiation, thereby promoting the differentiation of MSCs into cardiogenic cells. Despite the currently differentiated cardiogenic cells not displaying beating activity, the indispensable role of LncRNA-CIR6 in vitro in inducing MSC differentiation should not be overlooked. However, the specific mechanism governing this process remains unknown and requires further exploration in subsequent studies.

Existing research has underscored the potential of stem cell therapy to compensate for damaged cardiac muscle cells, emphasizing the crucial role of cardiac cell differentiation in replacing those affected by myocardial infarction [42]. In vitro, we validated the efficacy of LncRNA-CIR6 in promoting the differentiation of MSCs into cardiogenic cells. In murine models, we observed the positive impact of LncRNA-CIR6 in resisting myocardial infarction (MI). Particularly promising is the enhanced performance in heart function and the reduction in infarct size when LncRNA-CIR6 is combined with hUCMSCs, surpassing the outcomes of using LncRNA-CIR6 alone. This suggests that MSCs transfected with LncRNA-CIR6 may possess superior myocardial regeneration capabilities.

Our hypothesis posits that LncRNA-CIR6 may induce the differentiation of endogenous stem cells in the heart or the injected hUCMSCs into cardiogenic cells. Furthermore, studies have demonstrated that cardiac-inducing RNA 6 (CIR6) can induce the direct differentiation of murine fibroblasts into cardiomyocytes in vitro [43]. Therefore, we hypothesize that cardiogenic cells not only derive from the differentiation of MSCs induced by LncRNA-CIR6 but may also transform from myocardial fibroblasts under specific conditions. These regenerated cardiogenic cells play a role in resisting myocardial infarction, offering a potential pathway for heart regeneration. To validate these observations and speculations, further research is planned.

Additionally, we hypothesize that, besides their involvement in the differentiation into cardiogenic cells to replace infarcted cells, both LncRNA-CIR6 and MSCs play distinct roles in conferring cardioprotective effects. In recent years, LncRNAs have garnered significant attention due to their diverse functions and regulatory roles in cellular processes [44]. For instance, LncRNA-NRF has been identified as a regulator of cardiomyocyte death by targeting Mir-873 and RIPK1/RIPK3. The involvement of MSC-derived LncRNAs and exosomes in cardiac injury and repair is well-established [45]. Li et al. [46] elucidated the protective effect of bone marrow mesenchymal stem cell-derived exosomes on hypoxic-reperfusion injury in cardiomyoblasts through the HAND2-AS1/miR-17-5p/Mfn2 axis. Consequently, the possibility that MSC exosomes contribute to heart protection cannot be overlooked. Hence, the investigation into LncRNA-mediated regulation of cardiomyocyte regeneration and proliferation, aimed at enhancing cardiac function post-infarction, warrants further exploration.

In our bioinformatics database analysis, we precisely identified the location of the LncRNA-CIR6 transcript and noted its structural similarity to the transcript of n-sulfonyl glucose sulfate hydrolase. The database also predicted proteins interacting with LncRNA-CIR6, with CDK1 being one of them. CDK1 is a protein associated with cardiovascular disease and is involved in the proliferation and differentiation of cardiomyocytes [32]. Under the induction of LncRNA-CIR6, we observed an increased expression of CDK1 in MSCs. Moreover, inhibiting CDK1 made it challenging for MSCs to differentiate into cardiogenic cells. This suggests that CDK1 may play a crucial role in the induction of cardiogenic cell differentiation by LncRNA-CIR6.

CDK1, a cyclin-dependent kinase crucial for cell cycle regulation [47], orchestrates cell cycle progression by phosphorylating various substrates [48]. Within the Wnt signaling pathway, CDK1 exerts inhibitory effects through diverse mechanisms [49,50]. For instance, it phosphorylates LRP6, inhibiting its activity and preventing Wnt ligand binding and signal transduction [51]. Additionally, CDK1 can regulate the Wnt signaling pathway by phosphorylating β-catenin and other components [52]. It is noteworthy that previous studies have closely linked CDK1 expression to the proliferation and differentiation of myocardial cells [53], particularly exhibiting high activity during embryonic development to promote myocardial cell proliferation and differentiation [32,54].

Studies by Gao et al. propose that inhibiting β-catenin expression facilitates the cardiac differentiation of MSCs [55]. Zhang et al. observed that overexpressing miR-499 in rat BM-MSCs promotes heart-specific gene expression and activates the Wnt/β-catenin signaling pathway by altering the phosphorylated/dephosphorylated β-catenin ratio [56]. Furthermore, some research suggests that Wnt signaling pathway inhibitors enhance the differentiation of mesenchymal stem cells into cardiac progenitors in vitro. Inhibiting Wnt signaling also promotes the proliferation of human mesenchymal stem cells, improving cardiomyopathy in vivo [57].

Therefore, we hypothesize that LncRNA-CIR6 induces MSC differentiation into cardiogenic cells by regulating the Wnt signaling pathway through CDK1. However, there is still much to unravel regarding the precise mechanisms through which CDK1 regulates the Wnt signaling pathway and influences cardiomyocyte differentiation in MSCs. Further studies are essential to deepen our understanding of these intricate processes.

While this study has uncovered significant findings, it is crucial to candidly acknowledge certain limitations in our experimental approach. To delve deeper into the in vivo induction process of stem cells, employing injected hUMSCs cells labeled with red fluorescent enzymes is essential. The tracking of cell aggregation and induced differentiation in the body is pivotal. However, our laboratory currently lacks the necessary advanced imaging equipment, specifically the In Vivo Imaging System (IVIS system), and our proficiency in labeling hUMSCs cells with fluorescent enzymes is limited. Despite these constraints, we plan to collaborate with other universities in the future, leveraging their equipment and technical support to complete this pivotal experiment. Furthermore, to explore further the protective role of LncRNA-CIR6 in myocardial infarction and its underlying mechanisms, we have sent mice transfected with LncRNA-CIR6 for genetic sequencing to a biotechnology company. Although results are not yet available, we intend to persist in our research to gain a more in-depth understanding. These challenges and limitations will serve as motivation for us to enhance the quality of our experiments, collaborate with other research teams, and ensure that our research meets the highest scientific standards.

## 4. Materials and Methods

### 4.1. Animals

Sprague Dawley (SD) rats (100 ± 5 g) and C57BL/6 mice (20 ± 1 g) were used, provided by the Laboratory Animal Center of Yanbian University (License No. SYXK (Ji) 2020-0009).

All male animals were kept in a clean, well-ventilated, and well-lit environment with a temperature of 25–26 °C and a humidity of 70%. All animal experiments in this study were approved by the Institutional Animal Protection and Use Committee of Yanbian University (Approval No. YD20240124001) and followed the National Institutes of Health Guidelines for Experimental Animal Protection.

### 4.2. Isolation and Culture of Rat Bone Marrow Mesenchymal Stem Cells (BMSCs)

SD rats were sacrificed by cervical dislocation and then immersed in 75% ethanol for 5 min. The rats were placed in a supine position on a super-clean bench, and the bilateral femurs were removed under sterile conditions. The muscle tissue attached to the surface of the femur was cleanly removed (note to avoid damaging the epiphysis), and the femur was placed in a sterile PBS-filled petri dish for later use. Subsequently, the ends of the epiphysis were cut to expose the medullary cavity. A syringe was used to aspirate α-MEM complete medium (Containing 10% fetal bovine serum and 1% P/S, Shanghai XP Biomed Ltd., Shanghai, China) to flush the bone marrow cavity into a centrifuge tube, and the cell suspension was swirled to disperse the cells. Centrifugation was performed at 1000 rpm for 5 min at room temperature [58]. The supernatant was discarded, and the cells were resuspended in 10 mL complete medium to make a single-cell suspension. This suspension was then transferred to a 10 cm culture dish, gently shaken to mix, and placed in a cell incubator at 37 °C, 5% CO_2_, and saturated humidity. After 24 h, the α-MEM complete medium was replaced, and the liquid was subsequently replaced every 2 days. When the fusion rate reached 80% to 90%, a subculture was performed.

### 4.3. Plasmid

The plasmid containing the LncRNA-CIR6 sequence (Figure 4) was constructed by Hanbio Biotechnology Co., Ltd., Shanghai, China. The backbone vector containing the LncRNA-CIR6 DNA sequence was pCDNA3.1-CMV-MCS-EF1-ZsGreen-T2A-puro. In pCDNA3.1-CMV-MCS-EF1-ZsGreen-T2A-puro, CMV and EF1 are gene promoters. MCS is a multiple cloning site; PuroR is a resistance gene fragment, and ZsGreen is a green fluorescent protein gene. T2A is a self-cleaving peptide sequence. The LncRNA-CIR6 DNA is preferably inserted at the BamHI/EcoRI multiple cloning site of the backbone vector.

The schematic representation of pCDNA3.1-CMV-MCS-EF1-ZsGreen-T2A-puro is shown in (Figure 5), which was synthesized by Hanbio (Shanghai) Co., Ltd (Shanghai, China).

### 4.4. Transfect LncRNA-CIR6 into MSCs In Vitro

Human umbilical cord mesenchymal stem cells (hUCMSCs) were purchased from Shanghai Kanglang Biotechnology Co., Ltd. (Shanghai, China). BMSCs were isolated from the femur of SD rats. When the primary cells were cultured for 7 days and the cell confluence reached 70–80%, JetPRIME DNA and siRNA (Polyplus, Berkeley, CA, USA) transfection reagents were used to transfect the MSCs. After 9 days of transfection, immunofluorescence and flow cytometry were performed on the 10th day (Figure 6).

### 4.5. Immunofluorescence

MSCs were divided into Vehicle, LncRNA-CIR6, and Ro 3306 groups according to the experimental protocol (Figure 4). After discarding the old medium, transfected cells were fixed with 2 mL of 4% paraformaldehyde per well (10 min) and then removed. Subsequently, cells underwent three 5-min washes with ice-based PBS (1 mL per well). Following permeabilization with 0.1% PBS-Tween, an overnight incubation was carried out with the primary antibody at 4 °C, and with the secondary antibody, it was conducted at room temperature for 1 h away from light. After the washing steps, 10 μL of anti-fluorescence quenching sealing solution (Beyotime, Shanghai, China) containing DAPI was added for cover slipping. Finally, fluorescence was observed using a fluorescence inversion microscope (Thermo Fisher Scientific, Waltham, MA, USA).

Primary antibody: Anti-Cardiac Troponin T antibody [EPR20266] at a dilution of 1:500 (Abcam, Waltham, MA, USA); secondary antibody: Goat Anti-Rabbit IgG H&L (AlexaFluor^®^647) preadsorbed (ab150083) at a dilution of 1:2000 (Abcam, USA); Ro 3306: CDK1 inhibitor (APExBIO, Houston, TX, USA).

### 4.6. Flow Cytometry

MSCs were divided into LncRNA-CIR6 and Vehicle groups for transfection (Figure 4). On the 10th day post-transfection, cell suspensions were collected (Density: 3 × 10^5^ cells/mL) and centrifuged at room temperature for 5 min in a 15 mL centrifuge tube. The supernatant was discarded, and the cells were fixed with 4% paraformaldehyde for 15 min to prevent cell cross-linking. After centrifugation, the supernatant was discarded, and cells were resuspended in pre-chilled 90% methanol for 15 min, facilitating further cell fixation and transparency treatment. Following two washes with PBS and centrifugation, cells were resuspended in diluted primary antibody and incubated at 4 °C for 1 h. Subsequently, cells were washed twice with PBS and centrifugation and resuspended in diluted secondary antibody, followed by a 30-min incubation at 4 °C. After two additional washes, cells were resuspended in PBS and analyzed for all groups on a flow cytometer (BD Biosciences, Franklin Lakes, NJ, USA). The same primary antibody and secondary antibody in Section 4.5.

### 4.7. Western Blot

MSCs were transfected according to the LncRNA-CIR6 group and Vehicle group, as illustrated in Figure 4. On the tenth day after transfection, the protein was extracted from each group of MSCs, and the protein concentration was determined using the BCA protein concentration assay kit (Solarbio, Beijing, China). After quantification, the samples were mixed with an appropriate amount of protein tracer loading buffer (Proteintech, Hong Kong, China) and denatured. The proteins were separated by SDS-PAGE gel electrophoresis and then transferred to the PVDF membrane (Merck, Rahway, NJ, USA). The SDS-PAGE gel percentages were 12%, and blotting membranes were blocked with 5% skim milk for 1 h. Subsequently, the membranes were incubated with the corresponding primary antibodies overnight at 4 °C. Afterward, the membranes were incubated with the corresponding secondary antibody for 1 h at room temperature. Following exposure with the enhanced chemiluminescence kit (Merck, USA), the bands were processed using Image J software 1.8.0 (National Institutes of Health, Bethesda, MD, USA).

Primary antibodies: Anti-Cardiac Troponin T antibody [EPR20266] (ab209813) (Abcam, Waltham, MA, USA) at a dilution of 1:5000; Anti-CDK1 (phospho T161) + CDK2/CDK3 (phospho T160) antibody [EPR19546] (ab201008) at a dilution of 1:2000 (Abcam, USA); β-Actin Antibody (4967S) at a dilution of 1:1000 (CST, Danvers, MA, USA). Secondary antibody: Anti-rabbit IgG, HRP-linked Antibody (7074S) at a dilution of 1:1000 (CST, USA).

### 4.8. Experimental Protocols

The mice were randomly assigned to four groups: Sham group, MI group, LncRNA-CIR6 group, and LncRNA-CIR6 + huCMSCs group (Figure 6).

Sham group: Each mouse was injected with 100 μL of AAV9-cTnT-ZsGreen. After 3 weeks, a thoracotomy was performed without ligation, followed by the injection of 200 μL of normal saline one week later.

MI group: Each mouse received an injection of 100 μL of AAV9-cTnT-ZsGreen. After a 3-week period, a thoracotomy was performed to induce MI, followed by the injection of 200 μL of normal saline one week later.

LncRNA-CIR6 group: Each mouse was injected with 100 μL of AAV9-cTnT-LncRNA- CIR6-ZsGreen. After 3 weeks, a thoracotomy was performed to induce MI, followed by the injection of 200 μL of normal saline one week later.

LncRNA-CIR6 + huCMSCs group: Each mouse was injected with 100 μL of AAV9-cTnT- LncRNA-CIR6-ZsGreen. After 3 weeks, a thoracotomy was performed to induce MI, and one week later, 200 μL of hUCMSCs (injection volume: 5 × 10^5^ cells/100 μL) were administered.

### 4.9. Adeno-Associated Virus (AAV)-Infected Mice

C57BL/6J mice (male, 3–4 weeks) were used. Serotype 9 AAV vectors (AAV9) encoding LncRNA-CIR6 were provided by Hanheng Biotechnology (Shanghai, China). Mice were randomly chosen to receive a single-bolus tail vein injection of either AAV9 encoding AAV9-cTnT-LncRNA-CIR6-ZsGreen or AAV9-cTnT-ZsGreen at 1.5 × 10^12^ vg/mL 100 μL per mouse. After 3 weeks, the expression of NEU1 in the hearts of mice was measured by immunohistochemistry. (Figure 6).

### 4.10. “Heart Pop Out” Method of MI Model in Mice In Vivo

Adeno-associated virus (AAV)-infected mice, anesthetized with isoflurane (0.3 mL/min), were intubated using a laryngoscope.

After confirming deep anesthesia, the mouse was secured in the supine position on the operating table, and the left anterior chest surgical area was routinely prepared for skin disinfection.A 0.5 cm oblique incision was made approximately 1 cm from the left edge of the sternum. The chest wall muscles were bluntly separated layer by layer, and the chest cavity was rapidly opened through the third or fourth intercostal space.The anterior descending coronary artery was identified at the junction between the lower margin of the left atrial appendage and the conus pulmonalis. The artery was swiftly ligated using an 8-0 suture, positioned 1–2 mm from its root.After confirming the completion of ligation and the absence of bleeding, the heart was gently returned to the chest without suturing the ribs. The positions of the pectoralis minor and pectoralis major muscles were restored to cover the wound in the intercostal space. The skin was pinched, and the chest cavity was promptly squeezed to expel air. Subsequently, the skin was quickly sutured to prevent the re-entry of air and the potential development of pneumothorax.Mice were then placed on a 37 °C thermostatic heating pad, and their condition was closely monitored. Once fully awake and turned over, the mice were returned to their cages.

### 4.11. Echocardiography

Cardiac function was determined by echocardiography (VEVO 2100 imaging system, American VisualSonics company, Bothell, WA, USA). After alignment in the transverse B-mode with the papillary muscles, cardiac systole function was measured on M-mode images, and diastolic function was measured with pulsed wave (PW) DopplerIn brief; the mice were anesthetized with isoflurane (2% for induction and 1–1.5% for maintenance) mixed in 0.6 L/min O_2_ via a facemask. Left ventricular chamber size and wall thickness were measured in at least three beats from each projection and averaged. The interventricular septal thickness at diastole and systole (LVS,d and LVS,s, respectively), fractional shortening (LVFS), and ejection fraction (LVEF) of the left ventricle were measured.

Not only were echocardiograms performed on mice in each group one week before surgery, but also, after administering hUCMSCs or saline, observations were made for 9 days. Echocardiographic examinations of the mice’s hearts were then conducted on the 10th day (Figure 7).

### 4.12. Electrical Mapping

Anesthetized mice in each group underwent a thoracotomy, exposing the heart. A multi-channel electrode was attached to the free wall of the left ventricle. The multi-electrode array (MEA) consists of 32 electrodes with a diameter of 0.1 mm, arranged in a 6 × 6 grid (size: 6 mm × 6 mm). The electrode spacing is 1.2 mm. The 32 recording electrodes are connected to a 32-channel amplifier and data acquisition system (EMS64-USB-1002, MappingLab Ltd., Oxford, UK). The sampling frequency of each channel was set to 10 kHz, and the field potential recording corresponded to that of the reference electrode placed on the cardiac perfusion metal cannula.

MEA utilized EMapScope software EMS64-USB-1003 (MappingLab Limited, UK) to provide 32 channels of non-invasive synchronous recordings of cell field potential. The EMapScope software enables real-time monitoring of field potentials across all 32 channels, based on which the heart rate can be calculated. The activation time of each channel is calculated based on the relative delay of the first detected waveform in the recording array.

To generate an epicardial propagation map, the activation time of each channel is represented by a color code in a 6 × 6 grid (ranging mainly from red to blue), displaying the original recording array. This generates an epicardial propagation map for a clearer view of the conduction pattern and direction (with more red indicating faster conduction speeds). Through the analysis of EMapScope software, ventricular conduction times, Dispersion Absolutez (indicates the heterogeneity of ventricular conduction; the smaller the dispersion, the better the heart function) and average conduction velocity can be obtained.

### 4.13. Histological Analysis

After the completion of the final echocardiogram, the hearts of the mice in each group were cut along the plane of ligation, with the ligation point serving as the horizontal reference. The cardiac tissues below the ligation point were preserved, frozen in OCT compound, and sectioned using a microtome to obtain 10 μm-thick slices.

A portion of these slices was utilized for observing frozen cardiac tissues of mice in each group under a fluorescence microscope (Thermo Fisher Scientific, USA). Another set of slices was subjected to Masson staining. The frozen cardiac tissue slices from each group were stained using Masson’s Trichrome Staining Kit (Solarbio, China). Subsequently, the degree of fibrosis was assessed using an inverted microscope (Thermo Fisher Scientific, USA).

### 4.14. Evans Blue-TTC Staining

After administering hUCMSCs or saline to mice in each group, a 9-day observation period followed. On the 10th day, the mice were anesthetized, and their chests were opened. The ascending aorta of each was carefully clamped with hemostatic forceps, and 0.5% Evans blue dye (Solarbio, China) was slowly injected from the proximal end of the clamp. Rapid bluing of the mouse heart was observed, with no significant changes noted below the ligature line.

The heart was then extracted, swiftly cleaned in PBS, frozen at −80 °C, cut into 0.5 mm slices, dyed in a 2% TTC solution (Solarbio, China) at 37 °C for 15 min, and photographed. Three main colors are predominantly presented: blue, red, and white. The red areas represent ischemic but viable myocardial tissue, the white areas depict infarcted tissue, and the blue areas signify normal myocardial tissue.

### 4.15. LncRNA-CIR6 Bioinformatics and Statistical Analysis

Analyze the potential functions of LncRNA-CIR6, including its expression, subcellular localization, interacting miRNAs, or the transcription factors that regulate it, using Annolnc2 (http://annolnc.gao-lab.org/, accessed on 10 February 2023).

### 4.16. Statistical Analysis

The experimental data were statistically processed using GraphPad Prism 9.0 software. All data are presented as means ± S.D. Comparisons between two independent samples were performed using an unpaired *t*-test. Comparisons between multiple groups were conducted using one-way ANOVA and two-way ANOVA, with a significance level set at *p* < 0.05.

## Figures and Tables

**Figure 1 ijms-25-03466-f001:**
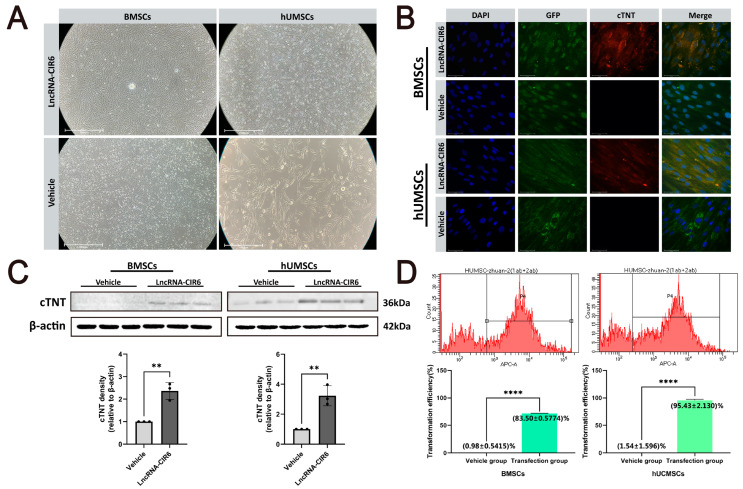
Effect of LncRNA-CIR6 on the induced differentiation of BMSCs and hUCMSCs into Ccardiomyogenic cells. (**A**) Inverted optical microscope images of MSCs transfected with empty plasmid or LncRNA-CIR6 plasmid on the 10th day (*n* = 3 per group); scale bar: 1000 μm. (**B**) Immunofluorescence map of MSCs transfected with empty plasmid or LncRNA-CIR6 plasmid on the 10th day (*n* = 3 per group); scale bar: 50 μm. (**C**) Western blot and quantification of cTNT in MSCs transfected with empty plasmid or LncRNA-CIR6 plasmid on the 10th day (*n* = 3 per group). (**D**) The fluorescence intensity of APC in MSCs transfected with empty plasmid or LncRNA-CIR6 plasmid on the 10th day (*n* = 3 per group). For all statistical plots, the data are presented as mean ± SD. ** *p* < 0.01, **** *p* < 0.0001 vs. Vehicle group (**C**,**D**). (Means ± SD, *n* = 3).

**Figure 2 ijms-25-03466-f002:**
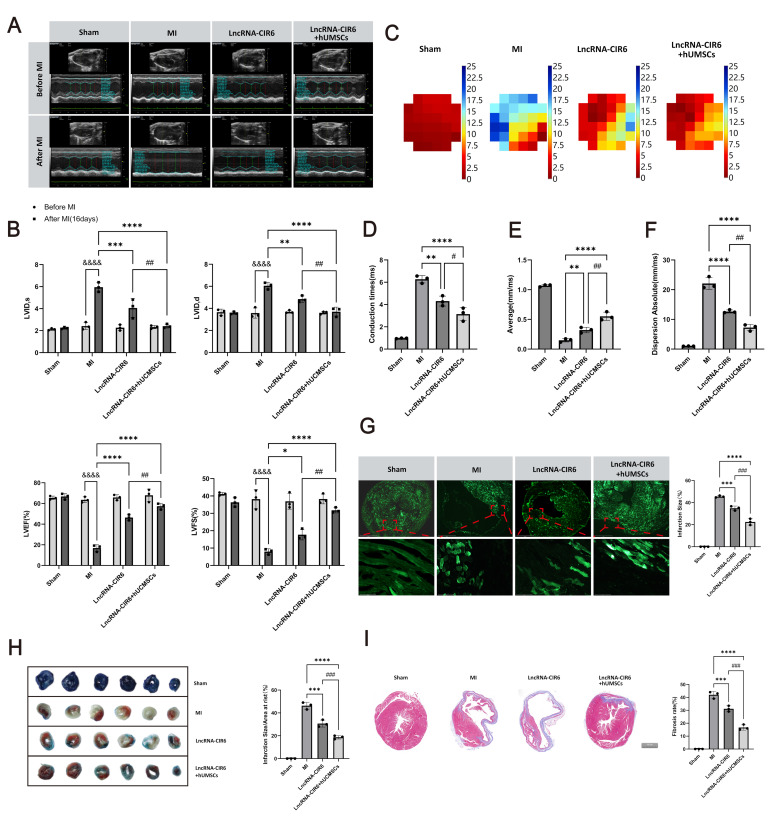
The effects of LncRNA-CIR6 or combined hUMSCs on heart function and cardioprotection in vivo in mice with myocardial infarction. (**A**) Representative echo image of M-mode after 17 days of MI was treated with LncRNA-CIR6 or combined hUMSCs (*n* = 3 per group). (**B**) Left ventricular EF (LVEF) and FS (LVFS), LVID,s and LVID,d assessed by echocardiography in mice (*n* = 3 per group). (**C**) Maps of ventricular conduction time in mice after 17 days of MI treated with LncRNA-CIR6 or combined hUMSCs (*n* = 3 per group). (**D**) Quantification of illustrated conduction time (*n* = 3 per group). (**E**) Quantification of average ventricular conduction velocity (*n* = 3 per group). (**F**) Quantification of dispersion absolute of ventricular conduction (*n* = 3 per group). (**G**) Fluorescence imaging of frozen mouse heart slices (scale bar: 1000 μm for upper panel; 75 μm for lower panel) for measuring infarct size after 17 days of MI was treated with LncRNA-CIR6 or combined hUMSCs (*n* = 3 per group). (**H**) Heart sections were stained with Evans blue-TTC, staining for infarct size; after 17 days of MI, they were treated with LncRNA-CIR6 or combined hUMSCs (*n* = 3 per group). (**I**) Heart sections were stained with Masson staining; after 17 days of MI, they were treated with LncRNA-CIR6 or combined hUMSCs (Scale bar: 500 μm) (*n* = 3 per group). For all statistical plots, the data are presented as mean ± SD. ^&&&&^
*p* < 0.0001 vs. Before MI group; * *p* < 0.05, ** *p* < 0.01, *** *p* < 0.001, **** *p* < 0.0001 vs. MI group; ^#^
*p* < 0.05, ^##^
*p* < 0.01, ^###^
*p* < 0.001, vs. LncRNA-CIR6 group (Means ± SD, *n* = 3).

**Figure 3 ijms-25-03466-f003:**
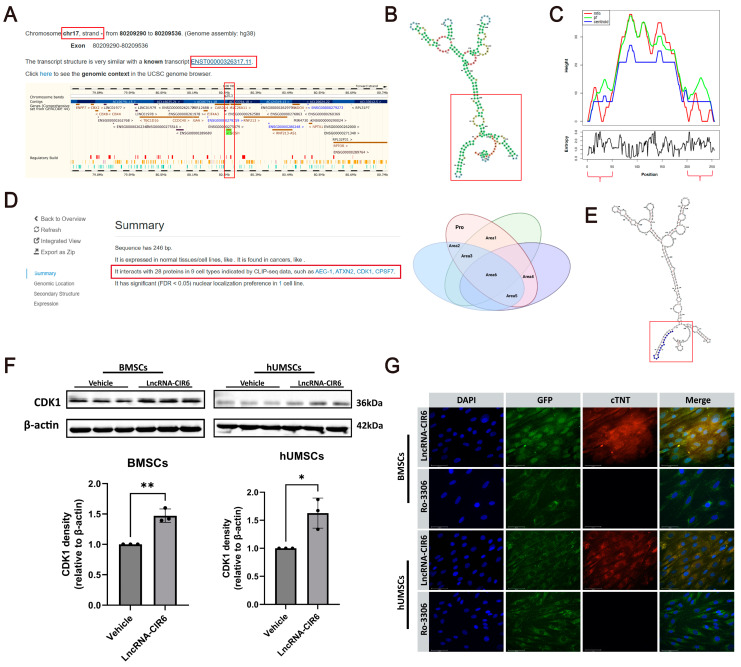
Prediction of LncRNA-CIR6 secondary structure and its protein interactions. (**A**) Transcription gene location of LncRNA-CIR6. (**B**) Predicted LncRNA-CIR6 secondary structure. (**C**) Mountain chart of LncRNA-CIR6 (LncRNA-CIR6 MFE (minimum free energy) structure), thermodynamic set of RNA structures, and centroid structure. (**D**) The Venn diagram below illustrates the interaction of 28 proteins with LncRNA-CIR6 (Area 1: Proteins binding to more than 10 binding sites on the secondary structure of LncRNA-CIR6. Area 2: Proteins with binding sites within the functional regions of LncRNA-CIR6. Area 3: Proteins with both more than 10 binding sites and located within the functional regions of LncRNA-CIR6. Area 4: Proteins associated with cardiovascular system diseases. Area 5: Proteins with binding sites within the functional region of LncRNA-CIR6 and associated with cardiovascular diseases. Area 6: Proteins with more than 10 binding sites, all in the functional region of LncRNA-CIR6, and these proteins are related to cardiovascular diseases). (**E**) CDK1 binding site on LncRNA-CIR6. (**F**) Western blot and quantification of CDK1 in MSCs transfected with empty plasmid or LncRNA-CIR6 plasmid on the 10th day (*n* = 3 per group). (**G**) Immunofluorescence map of MSCs transfected with LncRNA-CIR6 pretreated with/without Ro-3306 on the 10th day (*n* = 3 per group); Scale bar: 50 μm. For all statistical plots, the data are presented as mean ± SD.* *p* < 0.05, ** *p* < 0.01 vs. Vehicle group. (means ± SD, *n* = 3).

**Figure 4 ijms-25-03466-f004:**
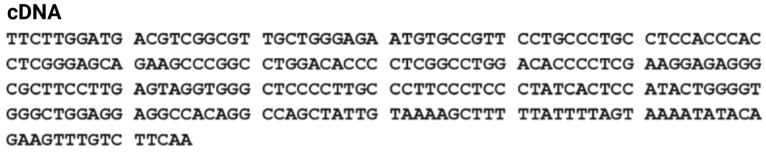
The cDNA sequence of LncRNA-CIR6.

**Figure 5 ijms-25-03466-f005:**
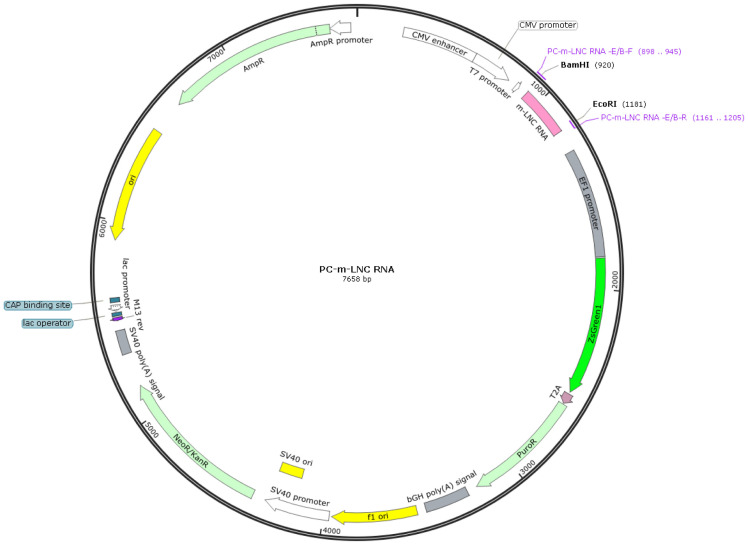
Construction of LncRNA-CIR6 plasmid.

**Figure 6 ijms-25-03466-f006:**
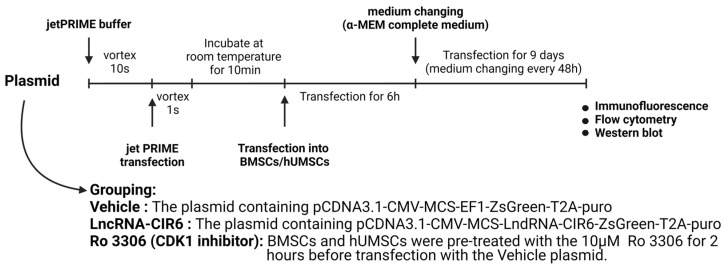
Experimental protocols and treatment reagents in vitro.

**Figure 7 ijms-25-03466-f007:**
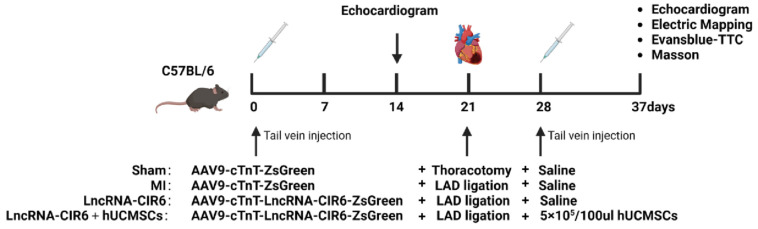
Experimental protocols and treatment reagents in vivo.

## Data Availability

Data are contained within the article.

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
