# Peer review of "Long Non-Coding RNA-Cardiac-Inducing RNA 6 Mediates Repair of Infarcted Hearts by Inducing Mesenchymal Stem Cell Differentiation into Cardiogenic Cells through Cyclin-Dependent Kinase 1"

_ijms, 2024, doi:10.3390/ijms25063466_

Round 1

Reviewer 1 Report

Comments and Suggestions for Authors

Major concerns:

1, In vitro experiments used mesenchymal stem cells. So, what's the rationale to use cTNT promoter for AAV transduction in vivo? Are there any cardiac stem cells that can utilize cTNT? If so, please isolate them and perform in vitro experiments.

2, IF staining shouldn't be used to quantitative purposes. WB and qPCR should be performed to support the conclusions.

3, A more comprehensive evaluation of the protective effects against MI should be performed.

Minor concerns:

1, More rationale should be given regarding why authors chose LncRNA-CIR6 in both introduction and discussion.

2, The figure quality can be improved. For example, the panel and font size of in Figure 4 are way too small. And they should be consistent.

3, Figure legends should give more details.

Comments on the Quality of English Language

English language must need to be improved.

Author Response

First of all, thank you so much for your comments. We have made extensive revisions and improvements throughout the entire article based on the valuable suggestions provided in the review. Each paragraph of the manuscript has been annotated accordingly. We would like to express our gratitude once again for the invaluable feedback you provided to us (We have replied to the questions in the attached document).

Reviewer 2 Report

Comments and Suggestions for Authors

The manuscript of Cui and collaborators aims to investigate the effect of LncRNA-CIR6 on mesenchymal stem cells (MSCs) differentiation into cardiac cell lineage in vitro and test its potential to regenerate myocardium after acute infarction in vivo.

LncRNA are critical epigenetic regulators of gene expression.

This paper suffers from important flaws in the methodology and also in the rational design of the research work, which undermine the validity and interpretability of the findings.

Specific criticisms:

- There are no data about endogenous expression and tissue specificity of the LncRNA-CIR6, not previously described in literature. No data about sequence conservation, crucial in this experimental setting.

The effects of LncRNA transfection in MSCs is not elucidated in details. The investigation of its ability to induce cardiogenic differentiation of MSCs from rat and human origin is limited by the sole examination of cTNT expression, and APC activation in BM-MCS. They do not provide details about cell morphology and expression of other cardiac markers, first of all sarcomeric proteins.

In addition to IF, an analysis of cardiac protein expression by western blot would be more informative.

Indeed, according to what reported in methods section, cells were not subjected to a specific protocol to cardiogenic differentiation in addition to LncRNA transfection. Is this molecule sufficient by its own to promote cell transdifferentiation?

- The in vivo experiments were conducted in mice, a different species from the MSCs tested in vitro. The LncRNA-CIR6 exhibited a remarkable cardioprotective effect after MI in treated animals in term of preserved function and myocardial tissue viability.

However, contrary to the in vitro approach, here the authors do not directly assess MSC transdifferentiation or tissue regeneration. Histological analysis to validate the effect of LncRNA-CIR6 after MI is notably absent, with no evidence provided for cardiomyocyte proliferation (i.e. incorporation of EdU, staining for Ki-67 or Aurora B) or decreased apoptosis, nor confirmation that tissue resident MSCs are involved. Therefore, it is not possible to establish the mechanism behind LncRNA-CIR6 cardioprotection.

- While it is conceivable that CDK1 could be a potential target for LncRNA-CIR6, it is important to note that CDK1 plays a crucial role in cell-cycle control. While it may be associated with the proliferation of spared cardiomyocytes at the infarct border zone, it cannot be considered a cardiogenic factor for mesenchymal stem cells. Moreover, it is conceivable that the inhibition of CDK1 activity leads to cell cycle arrest in G2/M, which could potentially hinder any potential myocardial recovery post myocardial infarction and induce apoptosis. Therefore, the interpretation of the results and conclusions of a direct involvement of CDK1 and MSCs activation lacks any scientific support.

- In the in vivo experiments, an AAV9 vector was utilized to facilitate the expression of LncRNA-CIR6. However, critical information regarding the titer of the purified vector preparation, the quantity of vector particles administered per animal, and the extent of myocardial transduction is absent.

Furthermore, systemic administration of AAV9 typically results in widespread transduction across various tissues and organs. Consequently, the overexpression of LncRNA-CIR6 outside of the cardiac tissue may potentially induce the secretion of circulating protective factors. It is not possible to exclude that such a mechanism could contribute to the observed protective effect following myocardial infarction.

- Notably, the number of animals included in each experimental group is not indicated. 

The bar plots should report the single measurements as dots.

- The resolution of the Figures is very low and is very difficult to read the details of the presented results.

-There are several writing flaws in the manuscript, particularly regarding lack of clarity in presenting the experimental procedures, which often lacks the required details, within Results section. Consequently, the comprehension of the presented findings results rather complicated in several parts.

Comments on the Quality of English Language

The overall quality of English language is acceptable, but the reading of the manuscript can be improved by a general revision.

Author Response

We have made extensive revisions and improvements throughout the entire article based on the valuable suggestions provided in the review. Each paragraph of the manuscript has been annotated accordingly. We would like to express our gratitude once again for the invaluable feedback you provided to us (We have replied to the questions in the attached document).

Reviewer 3 Report

Comments and Suggestions for Authors

I have reviewed the manuscript entitled “LncRNA-CIR6 Mediates Repair of Infarcted Hearts by Inducing MSCs Differentiation into Cardiogenic Cells through CDK1” submitted to IJMS. The authors touch on extremely important topics related to the implementation of new treatment methods designed to support the regenerative processes of the heart muscle. This study evaluates the impact of lncRNAs on abolishing the adverse effects of MI on cardiac performance by stimulating the differentiation of MSCs into cardiomyocytes. Below I present some comments and doubts that, once resolved, will complement this valuable manuscript.

·       In the introduction, the authors should also mention literature reports that did not indicate clear success after the implementation of various types of pluripotent or multipotent cells into the heart muscle.

·       Instead of examples relating to other diseases (lines 21-23), it will be better to cite works based on the heart muscle. You can also refer to clinical studies conducted on the heart.

·       In the description of the BMSCs isolation protocol, the authors provide very high centrifugation speeds, which may negatively affect the viability of the obtained cells. Please cite a literature source confirming the validity of using such centrifugation conditions.

·       In the same point, the authors mention the use of " α-MEM complete medium" - information regarding the supplementation of the culture medium used should be provided.

Author Response

(The authors gave the same response as above.)

Round 2

Reviewer 1 Report

Comments and Suggestions for Authors

My concerns were all solved. No more questions.

Comments on the Quality of English Language

Fine.

Author Response

Thank you sincerely for acknowledging our revised manuscript. Your invaluable suggestions have served as crucial guidance, and under your insightful direction, we have made adjustments to enhance the completeness and refinement of the article.

Your professional insights and patient guidance are indispensable assets to our team, and we feel honored to collaborate with you to elevate the quality of our work. We remain steadfast in our commitment to ensuring that our output aligns with your expectations and meets your specific requirements.

Guided by your expertise, we meticulously reviewed each section, implementing detailed modifications to ensure clarity and coherence in the structure, logic, and expression of the article.

Once again, we express our gratitude for your ongoing support and encouragement of our work. We eagerly anticipate the opportunity to deliver high-quality articles to you in the future.

Reviewer 2 Report

Comments and Suggestions for Authors

The authors revised the text of the manuscript, including details and new pictures, which improve the overall comprehension of the reported findings. There are, however, few persistent points of concern that I summarise below.

As previously underlined, I recognized that LncRNA-CIR6 may be of potential therapeutic interest; the reported results showed that this small molecule either alone and in conjunction with MCSs induces significant resistance of the myocardium to the damage due to myocardial infarction. 

The details provided now about AAV vector manufacturing, dosage of injection and evidence of robust expression in vivo of the transduced gene, corroborate the presented results. 

However, still there is no proof that the observed effect in treated animals is due to cardiogenic differentiation of endogenous or injected stem cells rather than protection from ischemia-induced apoptosis or to tissue regeneration through proliferation of the surviving cardiomyocytes. Unfortunately, the cardiac functional assessment was performed at a single time point after MI so it is not possible to infer if the myocardium survived the ischemic damage or regeneration occurred afterwards.

At least an additional echocardiography of control and treated animals should have been performed at an intermediate time point between LAD ligation and injection of huMCS; these data would have informed of the relative contribution of LncRNA-CIR6 and stem cells respectively. 

Similarly, affords to trace and/or detect injected MSC in treated animals would have rendered more solid the interpretation of the results.

Since injected cells are of human origin (how well are they tolerated in an allogeneic host?) their presence in the mouse histological samples can be evaluated using a human specific marker (antibody or probe). 

Alternatively, there are simple and inexpensive methods for non toxic, ex vivo labeling by lipophilic dyes staining cell membrane, such as DiO, DiI, DiD and DiR. Since these dyes are usually resistant to tissue fixation, transplanted cells can be tracked several days post injection in histologic tissue sections.

As a comment to the answer n.3 of the authors response, I just want to add that I personally used to routinely extract bone marrow cells from mouse femurs and to culture both hematopoietic and mesenchymal cells. I am aware that mouse MSCs rapidly undergo senescence, but nonetheless few animals are enough for starting and expanding a MSC culture.

Round 3

Reviewer 2 Report

Comments and Suggestions for Authors

The quality of the manuscript has improved after the revision and its principal findings more clearly delineated. In this present form it may worth publication